# Adding PD-1/PD-L1 Inhibitors to Chemotherapy for the First-Line Treatment of Extensive Stage Small Cell Lung Cancer (SCLC): A Meta-Analysis of Randomized Trials

**DOI:** 10.3390/cancers12092645

**Published:** 2020-09-16

**Authors:** Francesco Facchinetti, Massimo Di Maio, Marcello Tiseo

**Affiliations:** 1Predictive Biomarkers and Novel Therapeutc Strategies in Oncology, Inserm U981, Gustave Roussy Cancer Campus, Paris-Saclay University, 94800 Villejuif, France; FRANCESCO.FACCHINETTI@gustaveroussy.fr; 2Department of Oncology, University of Turin, at Ordine Mauriziano Hospital, 10128 Torino, Italy; 3Department of Medicine and Surgery, University of Parma, 43126 Parma, Italy; marcello.tiseo@unipr.it; 4Medical Oncology Unit, University Hospital of Parma, 43126 Parma, Italy

**Keywords:** small cell lung cancer (SCLC), immunotherapy, first-line chemotherapy, meta-analysis, PD-1/PD-L1

## Abstract

**Simple Summary:**

Treatment strategies in advanced, metastatic small cell lung cancer have been recently implemented by the combination of chemotherapy and immunotherapy. Nevertheless, the magnitude of survival benefit observed in clinical trials does not reproduce the major improvements observed in non-small cell lung cancer and other malignant diseases. By performing a systematic review and gathering the available data in a meta-analysis, we aim to compare the outcomes of patients treated with standard chemotherapy alone or with PD-1/PD-L1 inhibitors immunotherapy across clinical trials, in order to sustain treatment decisions. The addition of PD-1/PD-L1 inhibitors to standard chemotherapy improves all activity and efficacy outcomes, with a manageable safety profile. The benefit in overall survival is more evident if considering long-term analysis, compared to median estimations.

**Abstract:**

Survival outcomes in extensive-stage small cell lung cancer (ES SCLC) are dismal, with median overall survival (OS) less than 12 months. The combination of PD-1/PD-L1 immune checkpoint inhibitors (ICIs) with first-line platinum-etoposide chemotherapy has been recently evaluated in randomized clinical trials. We performed a systematic literature review through PubMed and conference proceedings. Randomized trials evaluating chemotherapy +/− PD-1/PD-L1 ICIs were included in the meta-analysis. Efficacy (OS), activity [progression-free survival (PFS) and objective response rate (ORR)] outcomes and toxicities were analyzed. For selected endpoints, we focused on patients’ subgroups (OS) and on landmark analyses (OS, PFS). Four randomized trials were identified; globally, 1553 patients were randomized to receive chemotherapy +/− PD-1/PD-L1 ICIs. Adding a PD-1/PD-L1 ICI to chemotherapy led to a significant benefit in OS [hazard ratio (HR) 0.76, 95% confidence interval (CI) 0.68–0.85, *p* < 0.00001), PFS [HR 0.75, 95% CI 0.68–0.84, *p* < 0.00001] and ORR [odds ratio 1.28, 95% CI 1.04–1.57, *p* = 0.02]. No unexpected toxicity emerged. At 12, 18, 24 months for OS, and at 12, 18 months for PFS, experimental arms retained significant improvement in event-free rates, with absolute gain of approximately 10% compared with standard treatment. Albeit the magnitude of the benefit is less impacting compared to other settings of immunotherapy, the addition of PD-1/PD-L1 ICIs to chemotherapy in ES SCLC provided significant improvements in survival outcomes with the known toxicity profile. Biomarkers predicting which patients are suitable to derive long-term benefits are eagerly awaited.

## 1. Introduction

Small cell lung cancer (SCLC) is a poorly differentiated neuroendocrine neoplasm accounting for about 15% of lung malignancies [1]. Its peculiar clinical features are represented by the strong etiopathogenic association with smoking history, the central thoracic localization and the rapid proliferation index, impacting on the highly symptomatic forms often observed already at disease diagnosis [2]. Synchronous and metachronous metastatic spread to lymph nodes, brain, liver and bone is frequent.

SCLC high proliferation index defines this tumor as chemo- and radio-sensitive, as rapid, relevant and symptomatic disease regression with chemotherapy +/− radiotherapy are normally achieved (the combination of the two treatments being administered with curative intent in limited-stage forms, LS SCLC). Nevertheless, in extensive-stage (i.e., metastatic, ES SCLC), disease progression is virtually unavoidable, with a median progression-free survival (PFS) shorter than six months [3]. Second-line treatment options are limited and globally disappointing, as chemo-resistance rapidly occurs, with response rates rarely exceeding 15–20% [4,5,6]. Median overall survival (OS) estimations in clinical trials of ES SCLC are therefore dismal, being approximately 10 months (since first-line initiation) [3]. 

For decades, platinum-etoposide has been the regimen of choice for SCLC patients, followed by second-line topotecan [4,5]. The outcomes of ED SCLC have been poorly improved by additional strategies such as prophylactic cranial irradiation (PCI, standard of care in LD) and consolidation thoracic radiotherapy, both of them controversial issues in ED SCLC management [7,8]. Additional strategies, such as maintenance therapies or the addition of anti-angiogenic agents to first-line treatment, failed to provide significant survival improvements [9,10].

SCLC immunogenicity has been evoked by (i) the expression of PD-1/PD-L1 detected in SCLC samples, especially in immune cells present in the stroma [11]; (ii) its strict correlation with smoking exposure, causing non-synonymous mutations therefore turning out in high tumor mutational burden (TMB) [12]; (iii) auto-antibodies cross-reacting with the neuronal antigen present in 16% of SCLC patients, leading to auto-immune paraneoplastic syndromes in approximately 5% of the cases [13,14]. To the detriment of initial enthusiasms [15], the first clinical proofs of PD-1/PD-L1 immune checkpoint inhibitors (ICIs, combined or not with CTLA-4 ICIs) have been disappointing in the pretreated setting of SCLC and as switch-maintenance strategies, frustrating, in addition, the role of PD-L1 as a biomarker [16,17,18,19]. Nevertheless, a subset of patients responded to the anti-PD-1 agent nivolumab or pembrolizumab when administered as the third or later treatment line (response rates 12–20%) and experienced very prolonged responses, as median durations of response were 17.9 months and not-reached (after 7.7 months of follow-up), respectively [20,21]. 

Similar to studies conducted in advanced NSCLC, anti-PD-1/PD-L1 ICIs have been evaluated in combination with platinum-etoposide chemotherapy for the first-line treatment of ED SCLC. Here, we provide a systematic review and meta-analysis of randomized clinical trials evaluating the addition of PD-1/PD-L1 ICIs to first-line chemotherapy in SCLC.

## 2. Results

### 2.1. Characteristics of the Trials 

The selection process of trials eligible for the meta-analysis is reported in Appendix A. In the search performed in June 2020, three trials were found eligible for inclusion [22,23,24]. A further publication reported updated toxicity and quality of life measures of one eligible trial [25]. In addition, four eligible trials were identified when searching the proceedings of the main International meetings, three of them representing presentations of trials already identified in PubMed, whereas one new study was identified [26,27,28,29].

Analysis of study quality was feasible for the three trials published in extenso [22,23,24] and only partially for the study still not published in extenso [29]. With this limitation, most of the evaluation criteria for the study quality checklist were fulfilled, with an overall quality score of B1 (sufficiently high quality to consider the risk of bias as low to moderate) for all included studies.

The main characteristics of the four trials included in the meta-analysis are reported in Table 1: one trial each with the PD-L1 ICIs atezolizumab (IMpower133), durvalumab (CASPIAN), or the PD-1 ICIs pembrolizumab (KEYNOTE-604) and nivolumab (ECOG-ACRIN EA5161). Besides the arms comparing chemotherapy alone or with durvalumab, CASPIAN trial included a third arm of chemotherapy + durvalumab + tremelimumab (CTLA-4 ICI). IMpower133 included a phase I part, representing a safety run-in period, followed by the phase III part of the study [22]. Disease responses were assessed according to RECIST version 1.1 in all the four trials.Standard chemotherapy in both standard and experimental arms included a platinum salt (either cisplatin or carboplatin, with the exception of IMpower133, where only carboplatin was allowed) administered on day 1 and etoposide administered on days 1–3, every three weeks. Slight differences between studies were recorded according to chemotherapy doses and the maximum number of cycles in the induction phase, as well as regarding the post-chemotherapy, maintenance phase with PD-1/PD-L1 ICIs (Table 1). Two studies were double-blind (IMpower133 and KEYNOTE-604) and two were open-label (CASPIAN and ECOG-ACRIN EA5161) with regard to PD-1/PD-L1 ICIs administration. 

The inclusion of patients with baseline brain metastases was allowed across all the trials, if clinically controlled and asymptomatic; in addition, two trials (IMpower133 and KEYNOTE-604) required brain metastases being treated with radiotherapy or equivalent before study enrolment. PCI was allowed, with the exception of the experimental arm of CASPIAN study; no information is available concerning its feasibility in ECOG-ACRIN EA5161. Thoracic radiation was explicitly not permitted only in IMpower133, while the continuation of trial treatment beyond radiological progression, in the case of clinical benefit, was openly admitted in all the three studies published *in extenso*. 

An update of IMpower133 was dedicated to toxicity and quality of life outcomes [25]. Nevertheless, as it reported adverse events (AEs) occurring in induction and maintenance phases of treatment separately, the total number of patients experiencing AEs during the whole study period could not be calculated, so toxicity data were obtained from previous communications. 

### 2.2. Patients’ Characteristics

Overall, 1553 patients were enrolled in the four trials included in the meta-analysis (intention-to-treat, ITT, population for efficacy analysis), 777 (50%) assigned to platinum-based chemotherapy + PD-1/PD-L1 ICIs, and 776 (50%) assigned to platinum-based chemotherapy alone. Main characteristics of the enrolled patients are described in Table 2. Patients were enrolled between June 2016 and December 2018. Median age was 62–65 years, and all patients had an ECOG performance status of 0 (*n* = 492, 31.7%) or 1 (1061, 68.3%). Male sex was predominant (*n* = 1000, 64.4%). Data regarding the platinum salt administered at study inclusion was available only for the three trials published *in extenso*: out of 1393 patients, 1122 (80.5%) and 271 (19.5%) received carboplatin and cisplatin, respectively. The number of patients with brain metastases was reported again only in published studies: overall, 145 out of 1393 (10.4%) patients had brain metastases at inclusion. Liver metastases were present in 548 out of 1393 patients (39.3%). Patients’ characteristics were well balanced between experimental and treatment arms (Table 2). The proportion of patients who received subsequent treatment lines was similar between the two arms, with slightly higher percentages in chemotherapy-only ones. 

### 2.3. Overall Survival

The addition of a PD-1/PD-L1 ICI to platinum-etoposide chemotherapy in patients with ES SCLC was associated with a statistically significant benefit in OS in the whole study ITT population (*n* = 1553), [hazard ratio (HR) 0.76, 95% confidence interval (CI) 0.68–0.85, *p* < 0.00001) (Figure 1). There was no evidence of asymmetry at the funnel plot (Appendix A). There was no evidence of significant heterogeneity among the four trials (*p* = 0.89, *I*² = 0%). In control and experimental arms, median OS ranged from 8.5 to 10.5 months and from 10.8 to 12.9 months, respectively (Table 1). The inclusion in the sensitivity analysis of the CASPIAN trial arm evaluating durvalumab and tremelimumab addition to chemotherapy (Appendix A), did not modify the estimations in favor of experimental treatments towards better OS (HR 0.77, 95% CI 0.69–0.86, *p* < 0.00001).

Landmark OS analyses at 12 and 18 months were available for three trials (IMpower133, CASPIAN and KEYNOTE-604), while CASPIAN and KEYNOTE-604 reported OS rates at 24 months. The 12-months OS rate in the control arms was 39–40% in the three studies, and was comprised from 45.1% to 52.8% in the experimental arms; 18-months OS rate rates ranged from 21% to 24.8% and from 30% to 34%, respectively. At 24-months, OS rate in control arms of CASPIAN and KEYNOTE-604 were respectively 14.4% and 11.2%, rising to 22% in the immunotherapy-containing regimens in both studies. Pooled analysis of the probability of being alive at the mentioned landmark time points is reported in Table 3. For all the three time-points mentioned, the probability to be alive was significantly in favor of the experimental arms, with a delta of +10.9% (95% CI from +5.6% to +16.1%, *p* = 0.0001) at 12 months, +9.4% (95% CI from +4.5% to +14.1%, *p* = 0.0002) at 18 months and +8.7% (95% CI from +1.8% to +15.4%, *p* = 0.013) at 24 months. 

In subgroup analyses (Figure 2), based on the data of the three trials published in extenso, the test for difference of treatment efficacy among the subgroups did not demonstrate a statistically significant interaction. 

### 2.4. Progression-Free Survival

In the whole population (*n* = 1553, data available for all the four eligible trials), the addition of a PD-1/PD-L1 ICI to platinum-etoposide chemotherapy in patients with ES SCLC was associated with a statistically significant benefit in progression-free survival (HR 0.75, 95% CI 0.68–0.84, *p* < 0.00001) (Figure 1). There was no evidence of significant heterogeneity among the four trials (*p* = 0.74, *I*² = 0%). In control and experimental arms, median PFS ranged from 4.3 to 5.4 months and from 4.8 to 5.5 months, respectively (Table 1). 

Landmark PFS analyses at 6 and 12 months were available for three trials (IMpower133, CASPIAN and KEYNOTE-604), while CASPIAN and KEYNOTE-604 reported PFS rates at 18 months. The 6-months PFS rate ranged from 22.4% to 45.8% in control arms and from 30.9% to 45.4% in experimental ones. At 12 months, only 5% of patients receiving chemotherapy only were progression-free across trials, while 12.6–17.9% had not progressed when a PD-1/PD-L1 ICI had been added to chemotherapy. At the 18-months landmark analysis, 2.1–3.4% and 10.8–13.9% of the patients in control and experimental arms, respectively, were still progression-free. Pooling PFS events (Table 3), PFS-rates at 12 and 18 months were significantly higher with PD-1/PD-L1 ICI addition to chemotherapy, with a delta of +3.6% (95% CI from −1.8% to +8.9%, *p* = 0.19) at 6 months, +10.9% (95% CI from +7.5% to +14.4%, *p* < 0.0001) at 12 months and +10.0% (95% CI from +6.1% to +13.9%, *p* < 0.0001) at 18 months.

### 2.5. Response Rate

Objective responses observed across trials are recapitulated in Figure 3. In the whole study population (*n* = 1553, data available for the four trials), the addition of a PD-1/PD-L1 ICI to platinum-etoposide chemotherapy in patients with ES SCLC was associated with a statistically significant increase in ORR [odds ratio (OR) 1.28, 95% CI 1.04–1.57, *p* = 0.02] (Figure 3). In the whole study population (*n* = 1553, data available for the four trials), a statistically significant increase in ORR [odds ratio (OR) 1.28, 95% CI 1.04–1.57, *p* = 0.02] was observed when PD-1/PD-L1 ICIs were added to platinum-etoposide chemotherapy in patients with ES SCLC (Figure 3).

Overall, ORR was equal to 59.0% (458/776) in the control arm vs. 64.7% (503/777) in the experimental arm. Response rates were higher in experimental arms in all trials but IMpower133, and there was a moderate, non-significant heterogeneity among trials (*I*² = 49%, *p* = 0.12). 

Focusing on complete responses (available in three trials), although uncommon, they too were obtained more frequently in experimental (2.4%) compared to standard treatment arms (0.9%) (Figure 3).

### 2.6. Toxicity

As expected, the addition of PD-1/PD-L1 ICs was accompanied with a numerical increase of AEs recorded through the three trials published *in extenso* (only grade 5, G5, treatment-related events could be extrapolated from ECOG-ACRIN EA5161) (Appendix A). Pooling patients having experienced AEs (Table 4), all-grade toxicities were more common in the experimental arms (OR 2.89, 95% CI 1.13–7.38, *p* = 0.03), while no significant difference was observed with regard to G3-4 events (OR 1.08, 95% CI 0.86–1.36, *p* = 0.51). Toxicities leading to the withdrawal of any treatment component were more common when PD-1/PD-L1 ICs were added to chemotherapy (OR 1.98, 95% CI1.36–2.89, *p* = 0.0004). No difference in G5 AEs emerged (Table 4). 

Focusing on immune-related AEs (irAEs), as expected they were more common in experimental arm (OR 3.18, 95% CI 2.35–4.29, *p* < 0.00001). When PD-1/PD-L1 ICIs were added to chemotherapy, the most frequently observed irAE, were dermatitis/rash (10.8%) and thyroid dysfunctions (16.3%). 

## 3. Discussion

Biological and clinical disease aggressiveness, as well as the lack of active treatment options, are the major responsible for the dismal survival outcomes of ES SCLC patients. Compared to the recent improvements in NSCLC management, the standard of care of SCLC has not been significantly modified since its establishment approximately 30 years ago. The results observed with ICIs in NSCLC since their introduction, from the advanced pre-treated setting to the earlier ones, are outstanding. Still with a less impressive magnitude of benefit, the combination of PD-1/PD-L1 ICIs to backbone first-line chemotherapy has a significant impact on all the observed outcomes, as shown in this meta-analysis.

Among the four trials adding PD-1/PD-L1 ICIs to platinum-etoposide therapy in ES SCLC, three were positive, satisfying their respective primary outcomes. Only KEYNOTE-604 (sharing independently-assessed PFS and OS as primary endpoints) resulted formally negative, as the improvement in OS obtained in the experimental arm containing pembrolizumab (HR 0.80, 95% 0.64–0.98, *p* = 0.0164) did not reach the established significance boundary of *p* = 0.0128 [24]. Albeit considered a secondary endpoint, OS was superior in the nivolumab arm of ECOG-ACRIN EA5161, compared to the chemotherapy-only one [29]. In all the trials, PFS was superior in the experimental compared to the standard treatment arms. 

The difference in median OS between standard and PD-1/PD-L1 ICI containing regimes ranged from 1.1 (KEYNOTE-604) to 2.8 (ECOG-ACRIN EA5161) months. The limited value of median estimations in assessing survival benefit in this setting is even more represented by the difference in median PFS, numerically favoring control arm (0.3 months) in CASPIAN and up to 0.9 months in favor of experimental regimens in IMpower133 and ECOG-ACRIN EA5161. As often observed in clinical trials evaluating ICIs in solid tumors, the improvements in survival outcomes can be more evident if looking at the tales of Kaplan-Meyer curves [30]. Indeed, when pooling the events of the published studies focusing on landmark analyses, the probability of being alive at 12, 18 and 24 months since treatment initiation in experimental arms was 50.2%, 32.0% and 22.3% respectively, exceeding of approximately 10% the corresponding rates in standard treatment (Table 3). With the exception of the 6-months timepoint (likely due to its early assessment), similar deltas between experimental and standard treatment arms were observed for PFS rates at 12 and 18 months (Table 3). A key element for clinical decision and for the future development of ICIs in this setting will be the definition of biomarkers predicting the patients suitable for long-term benefit. Thus far, PD-L1 has failed in this purpose and TMB evaluation, potentially linked to ICIs benefit when administered in pre-treated SCLC in retrospective evaluations [31,32], did not show predictive value in IMpower133 [24,26]. The recognition of diverse biological entities within SCLC, responsible for differential clinical behaviors and response to therapies, represent the most promising way to identify patients more likely to benefit from immunotherapy options [33].

Dealing with toxicity, irAEs occurring when PD-1/PD-L1 ICIs were added to chemotherapy had a mild impact on the overall toxicity, only slightly higher in the experimental arm (Table 4). No difference was recorded with regard to G3-4 AEs. Serious irAEs were observed in a minority of patients only, thus scaling down the putative risk of unleashing uncontrolled, previously subclinical autoimmune paraneoplastic syndromes, as seen with thymic malignancies [34]. 

Envisaging the translation of the observed results in the clinical practice, two major elements seem worthy of being mentioned. First, described trials only included ECOG PS 0-1 patients. Due to both comorbidities and disease burden nevertheless, ES SCLC patients present with poor PS at diagnosis in up to 30% of the cases [35,36,37]. Even lacking clear evidence of adding PD-1/PD-L1 agents to chemotherapy in ECOG PS 2-3 ES SCLC patients, this clinical behavior could anyway provide clinical benefit, as clinical improvements are usually rapidly observed after cytotoxic treatment onset, and immunotherapy (besides increasing the response rates, as seen), could therefore exert its effect on the medium-long term. Second, the inclusion of patients with brain metastases was allowed, but only in the case of their clinical stability and, in IMpower133 and KEYNOTE-604, before having been treated with radiotherapy. This is likely the reason why a smaller proportion of patients harbored brain disease in these clinical trials (10.4% overall, Table 2) compared to clinical practice (up to 24%) [38]. In daily routine moreover, patients with brain metastases are usually candidate to receive upfront chemotherapy (delaying radiotherapy), due to the good responses obtained by systemic therapy even at the intracranial level. Given the potentially detrimental effect of steroids on the activity of immunotherapy [39], patients undergoing high-dose steroids for symptomatic brain metastases could potentially initiate chemotherapy only, introducing PD-1/PD-L1 agents in case of intracranial disease response and steroids tapering. 

It could be argued that the patients who drive the largest benefits from immunotherapy addition are the same who could take advantage of their administration in the third or later treatment line [20,21]. This hypothesis represents speculation yet to be acknowledged, and moving PD-1/PD-L1 agents upfront may guarantee the full exertion of their synergy with chemotherapy. In addition, this strategy allows all the patients to be exposed to immunotherapy when they are still reasonably fit, as the recalcitrant and aggressive clinical behavior of SCLC progressively reduces patients receiving second and later treatment lines.

We recognize that a meta-analysis based on individual patient data (IPD) should be considered the optimal synthesis of evidence, because it could allow data verification, the potential update of follow-up compared to data reported in publications, better calculation and comparison of time to events, and more accurate study of treatment heterogeneity in patients’ subgroups [40]. However, considering that data of the eligible trials included in our systematic review are property of different sponsors (and that it is highly unlikely that all those data could be rapidly obtained and analyzed together), the present meta-analysis based on literature-based data has allowed a timely synthesis of the evidence available, and could be considered an acceptable surrogate of IPD meta-analysis.

## 4. Materials and Methods

### 4.1. Evidence Acquisition

#### 4.1.1. Identification of Eligible Trials and Evaluation of Study Quality

A search was performed in June 2020, to identify all randomized trials (both phase II and phase III) testing the addition of a single anti-PD-1 or anti-PD-L1 ICIs to first-line platinum-based chemotherapy in patients with ES SCLC. The literature search was performed using PubMed. The following key-words were used for the search: *(small cell lung cancer) AND (nivolumab OR pembrolizumab OR atezolizumab OR avelumab OR durvalumab) AND (random*)*. In addition, references of the identified articles were checked to identify further eligible trials. Furthermore, proceedings of the main International meetings [American Society of Clinical Oncology (ASCO) annual meeting, European Society of Medical Oncology (ESMO]) annual meeting, International Association for the Study of Lung Cancer (IASLC) World Conference on Lung Cancer], were searched from 2017 onwards for relevant abstracts. 

When more than one analysis was available for the same clinical trial, we included in the analysis the most recent information (corresponding to longer follow-up).

All the eligible trials were assessed for study quality and potential bias using a structured checklist based on the Method for Evaluating Research and Guideline Evidence (MERGE) criteria. In detail, the checklist considers the quality of randomization, blinding, outcome measures, measure assessment, arm comparability, loss to follow-up, and intention to treat analysis. This checklist allows to assign a quality score to each trial: A (low risk of bias), B1 (low to moderate risk of bias), B2 (moderate to high risk of bias), C (high risk of bias).

#### 4.1.2. Data Collection 

We performed a meta-analysis aggregating data obtained from study publications.

For each eligible trial, the following data were collected, if available: main inclusion criteria: patients’ age, ECOG performance status (PS), disease stage;details of treatment on study: platinum-based chemotherapy (drugs, doses and number of cycles), immune checkpoint inhibitor (drug, dose and duration of treatment);clinical trial methodology: primary endpoint, study hypothesis;patients’ accrual and follow-up: the date of start and date of the end of accrual; number of patients randomized to experimental arm, number of patients randomized to control arm, median follow-up;OS: number of deaths in each arm, median OS, HR with 95% CI, *p*-value; for each arm, probability of being alive at 12 months, 18 months, 24 months from randomization. Details (number of patients and deaths in each arm, HR with 95% CI) were collected also for selected subgroups: age (< 65y/≥ 65y), sex (males/females), ECOG PS (0/1), liver metastases (present/absent), brain metastases (present/absent), type of platinum administered (cisplatin/carboplatin).PFS: number of events in each arm, median PFS HR with 95% CI, *p*-value; for each arm, probability of being progression-free at 6 months, 12 months, 18 months from randomization.ORR and complete response rate: number of events in each arm.Toxicity: number of events of all-grades and G3–4 AEs; irAEs; AEs leading to the withdrawal of any treatment; G5 AEs; all-grades and G3–4 organ-specific irAEs.

#### 4.1.3. Statistical Methods

After data were abstracted, the analysis was performed with the Review Manager (RevMan (Computer program). Version 5.3. Copenhagen: The Nordic Cochrane Centre, The Cochrane Collaboration, 2014.) software. For all the trials included in the meta-analysis, efficacy (OS) and activity (PFS, ORR, complete response rate) were analyzed from all randomly assigned patients on an ITT basis. 

For both OS and PFS, the summary measure was the hazard ratio (with 95% CI). For response rate and toxicity, the summary measure was the odds ratio (with 95% CI). A fixed-effects model was applied, and all analyses were repeated also with a random-effects model to verify consistency. Statistical heterogeneity among the included studies was examined using the χ^2^ test and the I^2^ statistic. The latter statistic expresses the proportion of the total observed variability attributed to study heterogeneity. In order to assess publication bias, although its reliability is limited due to the low number of included studies, we examined funnel plot, with standard error of the log (Hazard Ratio) on the y-axis and the Hazard Ratio on the x-axis.

In one trial [23,27], a further experimental arm was reported, testing the addition of durvalumab and tremelimumab to chemotherapy. Although all the other comparisons included in the main analysis tested the addition of single-agent ICIs, we performed an exploratory analysis also adding this comparison. However, because that trial used the same control arm for both comparisons (durvalumab + chemotherapy vs. chemotherapy alone, and durvalumab + tremelimumab + chemotherapy vs. chemotherapy alone), the weight of each comparison was reduced according to a correction factor equal to the number of events actually observed in the trial, divided by the number of events taken into account in the analysis (where the control arm was counted twice). For both comparisons, this correction produced a prudential increase in the width of the CI for the estimated HR.

Subgroup analyses were performed testing the heterogeneity of treatment effect among subgroups, using an interaction test. 

For the calculation of the pooled probability of being event-free at prespecified time points (12, 18 and 24 months for OS; 6, 12 and 18 months for PFS), only trials displaying numbers of patients at risk at each defined landmark time were included in the pooled population; when not available, the probability of being event-free for each specific time points was inferred, with approximation, from Kaplan-Meier survival curves. 

Given the unavailability of standard errors/confidence intervals for the probability, in the pooled calculation the probability reported in each trial was weighted by the number of patients at risk. At each time point, the difference in probability between treatment arms, the confidence interval and the statistical significance were calculated using MedCalc [41].

## 5. Conclusions

In this meta-analysis, the addition of PD-1/PD-L1 ICIs to backbone platinum-etoposide chemotherapy in the first-line treatment of ES SCLC has been shown to be active, effective, and safe. Albeit the magnitude of the effect may be deemed relative if compared to other settings and other diseases (namely NSCLC), the OS survival rates at 12-, 18-, and 24-months (50.2%, 32.0%, 22.3%, respectively) sustain the introduction of these regimens in the standard of care of ES SCLC patients. Further steps in the comprehension of SCLC biology will hopefully shed light on which patients derive the largest benefits from chemo-immunotherapy combinations [42]. After decades, these latter represent the novel standard of care for the treatment of ES SCLC and should be considered the benchmark for the development of even better treatment strategies. 

## Figures and Tables

**Figure 1 cancers-12-02645-f001:**
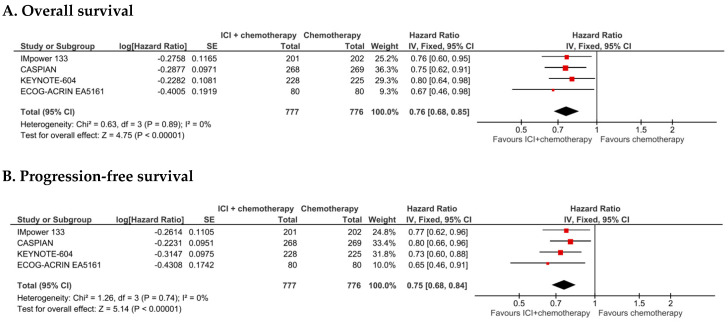
Overall survival (**A**) and progression-free survival (**B**) plots for the four clinical trials included in the meta-analysis, in the intention-to-treat populations. IV: Inverse variance; 95% CI: 95% confidence interval.

**Figure 2 cancers-12-02645-f002:**
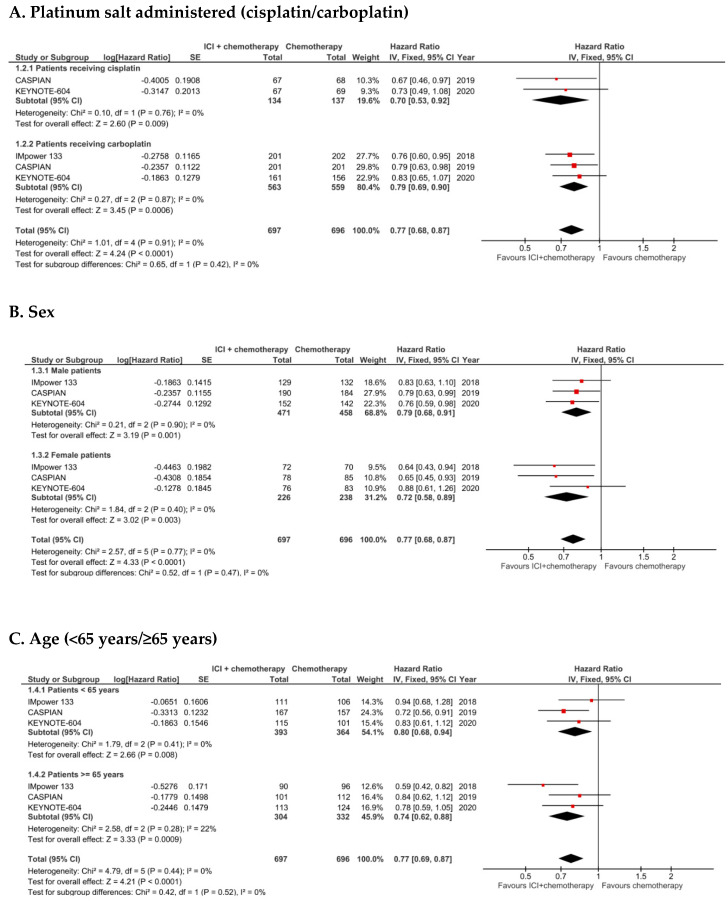
Overall survival. Subgroups analyses (**A**–**F**) for overall survival for the four clinical trials included in the meta-analysis, in the intention-to-treat populations. IV: Inverse variance; 95% CI: 95% confidence interval.

**Figure 3 cancers-12-02645-f003:**
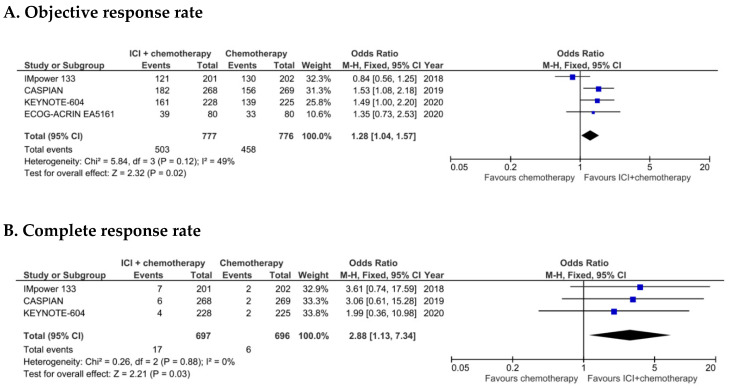
Objective response rate (**A**) and complete response rate (**B**) for the four clinical trials included in the meta-analysis, in the intention-to-treat populations. M-H: Mantel-Haenszel; 95% CI: 95% confidence interval.

**Table 1 cancers-12-02645-t001:** Characteristics of the trials included in the meta-analysis.

Trial	IMpower133	CASPIAN	KEYNOTE-604	ECOG-ACRIN EA5161
Reference, update	Horn New Engl J Med 2018 [22]	Paz-Ares Lancet 2019 [23]	Rudin J Clin Oncol 2020 [24]	Leal ASCO 2020 [29]
	Reck ESMO 2019 [26]	Paz-Ares ASCO 2020 [27]	Rudin ASCO 2020 [28]	
Study phase	I–III	III	III	II
Blinding	Double-blind	Open label	Double-blind	Open label
Treatment Platinum salt	Carboplatin AUC5	Carboplatin AUC5–6	Carboplatin AUC5	Carboplatin AUC5–6
		Cisplatin 75–80 mg/m^2^	Cisplatin 75 mg/m^2^	Cisplatin 75 mg/m^2^
Treatment Etoposide	100 mg/m^2^	80-100 mg/m^2^	100 mg/m^2^	100 mg/m^2^
Treatment Experimental arm	Atezolizumab 1200 mg	Durvalumab 1500 mg	Pembrolizumab 200 mg	Nivolumab 360 mg
Treatment Control arm	Placebo	/	Placebo	/
Treatment duration	4 cycles chemo	6 cycles chemo arm	4 cycles chemo	4 cycles chemo
	Maintenance atezo/placebo	4 cycles combo + maintenance durva q4w	Maintenance atezo/placebo up to 35 cycles	4 cycles combo + maintenance nivo 240 mg q2w
Primary endpoints	OS, PFS (IA)	OS	OS, PFS (BICR)	PFS
Randomized patients	403	537	453	160
Recruitment	June 2016–July 2017	March 2017–May 2018	May 2017–June 2018	May 2018–December 2018
Stratification factors	Sex, ECOG PS, brain mets	Type of platinum salt	Type of platinum salt, EGOG PS, LDH	Sex, LDH
Median follow-up (months)	22.9	25.1	21.6	NA
Median PFS experimental/standard arms (months)	5.2 (4.4–5.6)	5.1 (4.7–6.2)	4.8 (4.3–5.4)	5.5
	4.3 (4.2–4.5)	5.4 (4.8–6.2)	4.3 (4.2–4.5)	4.6
Median OS experimental/standard arms (months)	12.3 (10.8–15.8)	12.9 (11.3–14.7)	10.8 (9.2–12.9)	11.3
	10.3 (9.3–11.3)	10.5 (9.3–11.2)	9.7 (8.6–10.7)	8.5

Data between parenthesis represent 95% confidence intervals. AUC: Area under the curve; q4w: Every four weeks; q2w: Every two weeks; OS: Overall survival; PFS: progression-free survival; IA: Investigator-assessed; BICR: Blinded independent central review. ECOG PS: Eastern Cooperative Oncology Group performance status; PFS: Progression-free survival; OS: Overall survival; NA: Not available.

**Table 2 cancers-12-02645-t002:** Patients’ characteristics across trials and as gathered in the meta-analysis.

Patients’ Characteristics	IMpower133 [22,26]		CASPIAN [23,27]		KEYNOTE-604 [24,28]		ECOG-ACRIN EA5161 [29]		Total	
	Experimental	Control	Experimental	Control	Experimental	Control	Experimental	Control	Experimental	Control
	(*n* = 201) (%)	(*n* = 202) (%)	(*n* = 268) (%)	(*n* = 269) (%)	(*n* = 228) (%)	(*n* = 225) (%)	(*n* = 80) (%)	(*n* = 80) (%)	(*n* = 777) (%)	(*n* = 776) (%)
**Sex**										
Male	129 (64.2)	132 (65.3)	190 (70.9)	184 (68.4)	152 (66.7)	142 (63.1)	35 (43.7)	36 (45)	506 (65.1)	494 (63.7)
Female	72 (35.8)	70 (34.7)	78 (29.1)	85 (31.6)	76 (33.3)	83 (36.9)	45 (56.3)	44 (55)	271 (34.9)	282 (36.3)
**Age**										
<65 years	111 (55.2)	106 (52.5)	167 (62.3)	157 (58.4)	115 (50.4)	101 (44.9)	NA	NA	393 (56.4) *	364 (52.3) *
≥65 years	90 (44.8)	96 (47.5)	101 (37.7)	112 (41.6)	113 (50.6)	124 (55.1)	NA	NA	304 (43.6) *	332 (47.7) *
**ECOG PS**										
0	73 (36.3)	67 (33.2)	99 (36.9)	90 (33.5)	60 (26.3)	56 (24.9)	23 (28.7)	24 (30)	255 (32.8)	237 (30.5)
1	128 (63.7)	135 (66.8)	169 (63.1)	179 (66.5)	168 (73.7)	169 (75.1)	57 (71.3)	56 (70)	522 (67.2)	539 (69.5)
**Platinum salt**										
Carboplatin	201 (100)	202 (100)	201 (75)	201 (74.7)	161 (70.6)	156 (69.3)	NA	NA	563 (80.8) *	559 (80.3) *
Cisplatin	0	0	67 (25)	68 (25.3)	67 (29.4)	69 (30.7)	NA	NA	134 (19.2) *	137 (19.7) *
**Brain metastases**										
No	184 (91.5)	184 (91.1)	240 (89.6)	242 (90)	195 (85.5)	203 (90.2)	NA	NA	619 (88.8) *	629 (90.4) *
Yes	17 (8.5)	18 (8.9)	28 (10.4)	27 (10)	33 (14.5)	22 (9.8)	NA	NA	78 (11.2) *	67 (9.6) *
**Liver metastases**										
No	124 (61.7)	130 (63.4)	160 (59.7)	165 (61.3)	133 (58.3)	133 (59.1)	NA	NA	417 (59.8) *	428 (61.5) *
Yes	77 (38.3)	72 (35.6)	108 (40.3)	104 (38.7)	95 (41.7)	92 (40.9)	NA	NA	280 (40.2) *	268 (38.5) *
**Smoking status**										
Current	74 (36.8)	75 (37.1)	120 (44.8)	126 (46.8)	148 (64.9)	133 (59.1)	NA	NA	342 (49.1) *	334 (48.0) *
Former	118 (58.7)	124 (61.4)	126 (47.0)	128 (47.6)	72 (31.6)	84 (37.3)	NA	NA	316 (45.3) *	336 (48.3) *
Never	9 (4.5)	3 (1.5)	22 (8.2)	15 (5.6)	8 (3.5)	8 (3.6)	NA	NA	39 (5.6) *	26 (3.7) *

ECOG PS: Eastern Cooperative Oncology Group performance status. NA: Not avaliable. * Percentages are obtained out of a total of *n* = 697 and *n* = 696 patients in experimental and control arms, respectively, as information was not available regarding ECOG-ACRIN EA5161 trial.

**Table 3 cancers-12-02645-t003:** Pooled landmark analyses for OS and PFS.

Landmark Survival Analyses	Control Arm			Experimental Arm			Delta		
	Patients at risk	Probability	95% CI	Patients at risk	Probability	95% CI	%	95% CI	*p* value
12-months OS (3 trials) [26,27,28]	267	0.3932	0.3564–0.4312	335	0.5021	0.4634–0.5408	10.89%	5.58–16.11%	0.0001
18-months OS (3 trials) [26,27,28]	147	0.2265	0.1956–0.2615	206	0.3201	0.2847–0.3582	9.36%	4.50–14.16%	0.0002
24-months OS (2 trials) [27,28]	32	0.136	0.0963–0.1883	56	0.2228	0.1742–0.2807	8.68%	1.83–15.40%	0.0131
6-months PFS (3 trials) [26,27,28]	210	0.3508	0.3132–0.3911	253	0.3865	0.3496–0.4256	3.57%	−1.78–8.88%	0.1916
12-months PFS (3 trials) [26,27,28]	32	0.0523	0.0366–0.0738	98	0.1611	0.1334–0.1934	10.89%	7.48–14.36%	<0.0001
18-months PFS (2 trials) [27,28]	10	0.0301	0.0154–0.0564	50	0.1297	0.0988–0.1686	9.96%	6.10–13.93%	<0.0001

OS: Overall survival; PFS: Progression-free survival. 95% CI: 95% Confidence interval.

**Table 4 cancers-12-02645-t004:** Pooled analysis of toxicity data in the safety population of control and experimental arms.

Adverse Events	Control Arm Events/Patients	%	Experimental Arm Events/Patients	%	Odds Ratio (95% CI)	*p* Value
All adverse events	669/685	97.7	681/686	99.3	2.89 (1.13–7.38)	0.03
Grade 3-4 adverse events	458/685	66.9	470/686	68.5	1.08 (0.86–1.36)	0.51
AE leading to withdrawal of any treatment component	45/685	6.6	84/685	12.3	1.98 (1.36–2.89)	0.0004
Grade 5 adverse events	12/755	1.6	16/761	2.1	1.33 (0.63–2.84)	0.46
Immune-related AEs	78/685	11.4	190/685	27.7	3.18 (2.35–4.29)	<0.00001
Dermatitis/rash, all grades	36/685	5.3	74/686	10.8	2.25 (1.48–3.44)	0.0002
Dermatitis/rash, grade 3-4	0/685	0	7/686	1.0	8.09 (1.01–64.99)	0.05
Hepatitis, all grades	9/685	1.3	26/686	3.8	2.81 (1.34–5.90)	0.006
Hepatitis, grade 3-4	0/685	0	11/686	1.6	8.45 (1.55–46.12)	0.01
Hypothyroidism, all grades	8/685	1.7	72/686	10.5	9.90 (4.73–20.73)	<0.00001
Hypothyroidism, grade 3-4	0/685	0	0/686	0	NA	-
Hyperthyroidism, all grades	11/685	1.6	40/686	5.8	3.68 (1.89–7.15)	0.0001
Hyperthyroidism, grade 3-4	0/685	0	1/686	0.1	3.01 (0.12–74.37)	0.50
Pneumonitis, all grades	12/685	1.8	21/686	3.1	1.77 (0.86–3.63)	0.12
Pneumonitis, grade 3-4	3/685	0.4	6/686	0.9	1.86 (0.51–6.83)	0.35
Colitis, all grades	3/685	0.4	10/686	1.5	3.03 (0.90–10.19)	0.07
Colitis, grade 3-4	2/685	0.3	4/686	0.6	1.67 (0.40–7.00)	0.49
Adrenal insufficiency, all grades	4/685	0.6	3/686	0.4	0.80 (0.21–2.97)	0.73
Adrenal insufficiency, grade 3-4	0/685	0	2/686	0.3	5.05 (0.24–105.69)	0.30
Type I diabetes, all grades	0/685	0	6/686	0.9	5.04 (0.87–29.13)	0.07
Type I diabetes, grade 3-4	0/685	0	5/686	0.7	6.07 (0.73–50.57)	0.10

G3-4 irAEs were rare, never exceeding 1%. AEs: Adverse events; 95% CI: 95% confidence interval.

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
