# Peer review of "Adding PD-1/PD-L1 Inhibitors to Chemotherapy for the First-Line Treatment of Extensive Stage Small Cell Lung Cancer (SCLC): A Meta-Analysis of Randomized Trials"

_cancers, 2020, doi:10.3390/cancers12092645_

Round 1

Reviewer 1 Report

In this study, Facchinetti and colleagues examined the benefit of adding PD-1/PD-L1 blockades to chemotherapy as the first-line treatment of ES-SCLC by a systematic review and meta-analysis. In general, meta-analysis is performed to provide evidence or new insights to a clear question that might not be fully answered in individual studies; or to resolve potentially conflicting results from individual reports by systematically assessing data from multiple cohorts. However, in this manuscript, both overall survival and PFS are significantly higher in the experimental arm in all the individual trials included. It is missing the key question that the authors aim to address, other than simply summarizing/reviewing literature on the well-known fact that adding PD-1/PD-L1 blockades to chemotherapy improves patient outcome in ES-SCLC. Surely, when you when you pull the results together, you will get a statistical significance in OS/PFS in experimental group. Most importantly, it fell short in providing new information than what we already knew regarding OS and PFS, likewise in the subgroup analysis too.

In addition, it is mentioned in the method part, both the fixed- and random-effects model were used. But it seems to be only data from fixed-effects were shown. The authors should clearly state the reason/circumstances why one model was chosen over the other in this datasets.

More information should be included regarding the trials in table 2, such as the median follow-up, median months of overall survival and progression survival. Also, the survival rates of indicated time, such as 1 year.

There were typos and grammar errors throughout the manuscript, as well as missing references in the introduction.

Author Response

In this study, Facchinetti and colleagues examined the benefit of adding PD-1/PD-L1 blockades to chemotherapy as the first-line treatment of ES-SCLC by a systematic review and meta-analysis. In general, meta-analysis is performed to provide evidence or new insights to a clear question that might not be fully answered in individual studies; or to resolve potentially conflicting results from individual reports by systematically assessing data from multiple cohorts. However, in this manuscript, both overall survival and PFS are significantly higher in the experimental arm in all the individual trials included. It is missing the key question that the authors aim to address, other than simply summarizing/reviewing literature on the well-known fact that adding PD-1/PD-L1 blockades to chemotherapy improves patient outcome in ES-SCLC. Surely, when you when you pull the results together, you will get a statistical significance in OS/PFS in experimental group. Most importantly, it fell short in providing new information than what we already knew regarding OS and PFS, likewise in the subgroup analysis too.

We thank the Reviewer for the comment. Indeed, among the four studies included in the meta-analysis, only two (Impower133 and CASPIAN) are formally positive for engendering survival benefit in a context of phase III trials. In fact, KEYNOTE-604 is formally negative in overall survival and ECOG-ACRIN EA5161 is a phase II trial, with PFS as the primary endpoint. Moreover, individual trials, still positive, showed an improvement in survival outcomes whose magnitude is not comparable to the benefit observed in NSCLC, questioning the real indication of PD-1/PD-L1 blockers in addition to chemotherapy in the clinical practice of advanced SCLC. We pooled trials results in order to provide global data on this strategy, and trying to find some subgroup of interest with more statistical power compared to single trials.

Our analysis shows that there is no subgroup of patients more suitable (or less suitable) to benefit from PD-1/PD-L1 inhibitors addition. Although we recognize that these subgroup analyses are exploratory and that interaction test has a limited statistical power, we do not feel this is a limitation to our study, but a description of the clinical evidence in this setting. In addition, the meta-analysis allowed to have a global view of the landmark survivals obtained with this strategy, as we believe that these could represent more significant readouts that median survival estimations in the interpretation of the evidence and in the discussion between physicians and patients about treatment benefit/risk ratio.

In addition, it is mentioned in the method part, both the fixed- and random-effects model were used. But it seems to be only data from fixed-effects were shown. The authors should clearly state the reason/circumstances why one model was chosen over the other in this datasets.

We thank the Reviewer for this comment. We decide to present only data from fixed-effects model because, for both overall survival and progression-free survival, we did not find significant heterogeneity among the trials included (and the similar mechanism of action of the different drugs included allows to suppose the absence of relevant heterogeneity). In the absence of heterogeneity, we believe that fixed-effects model produces a good summary estimate of the results. Following Reviewer’s comment, we added a sentence in the Methods to justify the decision of presenting only results from fixed-effects models. However, for the Reviewer, in the Table below we present the results of the random-effects model, for all the endpoints of efficacy included in the analysis. Results were identical for overall survival and progression-free survival, and very similar for objective response rate and complete response rate:

Endpoint

Heterogeneity (I2)

Fixed-effects model

Random-effects model

Hazard Ratio (95% CI)

Hazard Ratio (95% CI)

Overall survival

0% (p=0.89)

0.76 (0.68 – 0.85)

0.76 (0.68 – 0.85)

Progression-free survival

0% (p=0.74)

0.75 (0.68 – 0.84)

0.75 (0.68 – 0.84)

Odds Ratio (95% CI)

Odds Ratio (95% CI)

Objective response rate

49% (p=0.12)

1.28 (1.04 – 1.57)

1.27 (0.94 – 1.71)

Complete response rate

0% (p=0.88)

2.88 (1.13 – 7.34)

2.85 (1.11 – 7.30)

More information should be included regarding the trials in table 2, such as the median follow-up, median months of overall survival and progression survival. Also, the survival rates of indicated time, such as 1 year.

We agree with the Reviewer on this point and we have enriched Table 1 with useful information. We detailed landmark PFS and OS in the individual studies in the text.

There were typos and grammar errors throughout the manuscript, as well as missing references in the introduction.

We thank the Reviewer for having signaled this point. We have read again carefully the manuscript and provided all corrections. The introduction accounts already 21 reference and all of them seem appropriate to us. We will be eager to include suggested references.

Reviewer 2 Report

The study by Facchinetti et al performs a systematic literature review through PubMed and conference proceedings to evaluate the benefits of including ICI to systemic therapies in SCLC. Efficacy (OS), activity [progression-free survival (PFS) and objective response rate (ORR)] outcomes and toxicities were analyzed in four trials encompassing 1553 patients. Adding a PD-1/PD-L1 ICI to chemotherapy led to a significant benefit in OS [hazard ratio (HR) 0.76, 95% confidence interval (CI) 0.68 – 0.85, p < 0.00001), PFS [HR 0.75, 95% CI 0.68 – 0.84, p < 0.00001] and ORR [odds ratio 1.28, 95% CI 1.04 – 1.57, p = 0.02]. No unexpected toxicity emerged. At 12, 18, 24 months for OS, and at 12, 18 months for PFS, experimental arms retained significant improvement in event-free rates, with absolute gain of approximately 10% compared with standard treatment. Authors suggest that the chemo-immunotherapy should be regarded as a new benchmark for comparing future therapies in SCLC.

Overall the study is significant and the paper is well written and the analyses are appropriate for the study. There are no major concerns from this reviewer regarding the publication of this study.

Minor comments:

  • It will be interesting to comment on why in the earlier trial by Horn (2018) the objective response rate (Figure 3) trended in the wrong direction (Odds ratio 0.84). Could that be due to a different patient demographic or specific treatment protocol?
  • References to the studies in the tables are based on the authors, but in the text they are usually by the trial names. This makes tracking a bit difficult.
  • Some abbreviations, such as AE and ir-AE are not defined as they appear in the text.
  • The use of the term “unrevealing” in the Discussion appears incorrect.
  • In the Discussion, authors state that the median PFS favors the control arm (0.3 months) in the CASPIAN trial. This appears to be contradicting the figures?

Author Response

The study by Facchinetti et al performs a systematic literature review through PubMed and conference proceedings to evaluate the benefits of including ICI to systemic therapies in SCLC. Efficacy (OS), activity [progression-free survival (PFS) and objective response rate (ORR)] outcomes and toxicities were analyzed in four trials encompassing 1553 patients. Adding a PD-1/PD-L1 ICI to chemotherapy led to a significant benefit in OS [hazard ratio (HR) 0.76, 95% confidence interval (CI) 0.68 – 0.85, p < 0.00001), PFS [HR 0.75, 95% CI 0.68 – 0.84, p < 0.00001] and ORR [odds ratio 1.28, 95% CI 1.04 – 1.57, p = 0.02]. No unexpected toxicity emerged. At 12, 18, 24 months for OS, and at 12, 18 months for PFS, experimental arms retained significant improvement in event-free rates, with absolute gain of approximately 10% compared with standard treatment. Authors suggest that the chemo-immunotherapy should be regarded as a new benchmark for comparing future therapies in SCLC.

Overall the study is significant and the paper is well written and the analyses are appropriate for the study. There are no major concerns from this reviewer regarding the publication of this study.

Minor comments:

It will be interesting to comment on why in the earlier trial by Horn (2018) the objective response rate (Figure 3) trended in the wrong direction (Odds ratio 0.84). Could that be due to a different patient demographic or specific treatment protocol?

We agree with Reviewer’s point. Indeed, we shared the same impressions while examining the data. Patients’ population included in IMpower133 did not differ significantly from other trials (ECOG PS 0-1, brain metastases allowed only if stable and pre-treated). IMpower133 was the only trial to allow carboplatin only in association to etoposide as a backbone chemotherapy, but as this happened both in the experimental and in the control arms, we do not think had a role in impacting on ORR in an anti-intuitive way. We can envisage that these data had not a true clinical explanation, but they rather are a consequence of the chance.

References to the studies in the tables are based on the authors, but in the text they are usually by the trial names. This makes tracking a bit difficult.

We thank the reviewer for this helpful suggestion. Indeed, the first line of Tables contains the trial names. We have made this line bigger, in order to make it more accessible to readers. In addition, we have provided the names of the trials, instead of the references, in figures and supplementary figures too.

Some abbreviations, such as AE and ir-AE are not defined as they appear in the text.

The use of the term “unrevealing” in the Discussion appears incorrect.

We agree with these two points and we have provided the respective corrections.

In the Discussion, authors state that the median PFS favors the control arm (0.3 months) in the CASPIAN trial. This appears to be contradicting the figures?

The comment of the Reviewer is really suitable to assessing the meaning of median estimations of survival outcomes in immunotherapy trials, such as the present ones. Median PFS slightly favors control arm in CASPIAN study, as reported by the data here presented of ASCO 2020 update. Nevertheless, the statistical approach, that takes into consideration the Kaplan-Meier curves in their global through a log-rank test, defined that the experimental arm was significantly superior to the control one (mainly thanks to the long-term outcomes represented by the tails of the curves).

Reviewer 3 Report

Important Meta-analysis adding data to promote ICIs in combination with standard double drug Cx to evidence level 1 in SCLC (ED).

Mayor criticism:

Materials:

  • "After data were abstracted, analysis was performed..." Make clear whether this is a meta-analysis based on the full publications or on all single patients files/data sets provided by the investigators of the trials. In case it was a publication based and not a single pt data based analysis discuss the possible shortcomings of the publication based approach for meta-analyses.
  • Is one of the authors a qualified biostatistician? Please state in revised paper.

Minor criticism:

Intro:

  • change "etiopathogenetic" into "etiopathogenic". It refers to genesis not to genetics.
  • "lack of solid second-line options" ...Be more cautious and quote 2nd line topotecan.

Discussion:

  • discuss the lack of effects in pts with brain metastasis in more detail

Author Response

Important Meta-analysis adding data to promote ICIs in combination with standard double drug Cx to evidence level 1 in SCLC (ED).

Mayor criticism:

Materials:

  • "After data were abstracted, analysis was performed..." Make clear whether this is a meta-analysis based on the full publications or on all single patients files/data sets provided by the investigators of the trials. In case it was a publication based and not a single pt data based analysis discuss the possible shortcomings of the publication based approach for meta-analyses.

We agree with Reviewer’s suggestion. Indeed, we have now stated more clearly that this is a publication-based and not an individual patient (IPD) data meta-analysis. We added a sentence in the Discussion about the limitation of literature-based meta-analysis compared to IPD meta-analyses: “We recognize that a meta-analysis based on IPD should be considered the optimal synthesis of evidence, because it could allow data verification, potential update of follow-up compared to data reported in publications, better calculation and comparison of time to events, and more accurate study of treatment heterogeneity in patients’ subgroups [Piedbois P, Buyse M. Meta-analyses based on abstracted data: a step in the right direction, but only a first step. J Clin Oncol. 2004 Oct 1;22(19):3839-41]. However, considering that data of the eligible trials included in our systematic review are property of different sponsors (and that it is highly unlikely that all those data could be rapidly obtained and  analyzed together), the present meta-analysis based on literature-based data has allowed a timely synthesis of the evidence available, and could be considered an acceptable surrogate of IPD meta-analysis.”

  • Is one of the authors a qualified biostatistician? Please state in revised paper.

We thank the Reviewer for this comment, of course a biostatistical expertise is fundamental for correctly conducting and interpreting a meta-analysis.However,  Prof. Di Maiohas a large experience in methodology of clinical research, with special regards to meta-analyses. He has performed a large number of meta-analyses, both individual patient data meta-analyses:

  • Di Maio M, et al.J Clin Oncol 2007; 25: 1377-1382
  • Di Maio M, et al. J Clin Oncol. 2009 Apr 10;27(11):1836-43.
  • Rossi A, et al. J Clin Oncol. 2012 May 10;30(14):1692-8.
  • Rossi A, et al. Lancet Oncol. 2014 Oct;15(11):1254-62.
  • Cremolini C et al, J Clin Oncol. 2020 Aug 20:JCO2001225. doi: 10.1200/JCO.20.01225.

and literature-based meta-analyses: 

  • Tucci M, et al. Eur Urol. 2016 Apr;69(4):563-73.
  • Ardizzoni A, et al. Lung Cancer. 2016 Oct;100:30-7.
  • Cremolini C et al..Cancer Res Treat. 2017 Jul;49(3):834-845.
  • Di Maio M et al. Critical Reviews in Oncology / Hematology , 2018, Volume 124 , 21 – 28;
  • Addeo A et al. Front Oncol. 2019 Apr 16;9:264. doi: 10.3389/fonc.2019.00264; .

We have stated this in the manuscript.

Minor criticism:

Intro:

  • change "etiopathogenetic" into "etiopathogenic". It refers to genesis not to genetics.
  • "lack of solid second-line options" ...Be more cautious and quote 2nd line topotecan.

We agree with both of the points and we have provided the respective changes.

Discussion:

  • discuss the lack of effects in pts with brain metastasis in more detail

We thank the reviewer for this suggestion. Indeed, we cannot state that patients with brain metastases do not benefit from PD-1/PD-L1 inhibitors addition to chemotherapy (the interaction test, with all its limitations, is formally negative). Of course, the number of patients with brain metastases included in the studies was limited, and only in IMpower133 this represented a stratification factor. Moreover, as seen if Figure 2F, the 95% confidence interval of patients without brain metastases (HR 0.75, 95% CI 0.66-0.85) overlaps with the one of patients with brain metastases (HR 1.00, 95% 0.69-1.44), precluding any definitive statement on patients with brain metastases compared with the ones without. Beyond statistical issues, the presence of brain metastases and the need for corticosteroids is of course a clinical problem, challenging the applicability of the addition of immunotherapy to chemotherapy in clinical practice, if we consider that patients with brain metastases are by far more common than the small proportion included in these trials.

Round 2

Reviewer 1 Report

  1. Given that SCLC is a completely different disease comparing to NSCLC, it seems to me very difficult to justify the authors' underlying reason/statements regarding the response to ICBs +chemotherapy: "magnitude is not comparable to the benefit observed in NSCLC, questioning the real indication of PD-1/PD-L1 blockers in addition to chemotherapy in the clinical practice of advanced SCLC." 
  2. it is true that one purpose of meta-analysis is to provide a global view on the topic of your interest, but most important it should also provide some new insights through statistical analysis. 
  3. the authors stated that the reasons why they choose fixed-effect model due to no significant heterogeneity in the included studies. could the authors provide evidence/analysis to support this? or have I missed it? also related, there is no analysis on quality of the study, publication bias and sensitivity analysis.